# TCAN: An Asymmetry Modeling Network for Time Series Forecasting

## Abstract

Most existing time series forecasting methods assume shared statistical consistencies across variables, such as periodicity. This assumption enforces symmetric modeling with shared encoders, yet real-world datasets often reveal distinct primary cycles for different variables. To address this gap, we introduce the Temporal Convolutional Association Block (TCAB), a flexible temporal convolution module that combines the strengths of attention and convolution to enable efficient asymmetric modeling of temporal and causal relationships. TCAB performs patch-wise equivalent sequence modeling by replacing attention score computation with learnable weights while preserving relative positional information. Building on TCAB, we propose the Temporal Convolutional Association Network (TCAN), a framework designed to capture asymmetric long-term dependencies and causal relationships across variables and patches. Extensive experiments on seven real-world datasets demonstrate that TCAN consistently outperforms state-of-the-art methods, validating the effectiveness of TCAB and providing a robust solution for efficient asymmetric modeling in multivariate time series forecasting. The code is available at https://anonymous.4open.science/r/TCAN-8F21.

## 1 Introduction

Time series forecasting (TSF) has attracted significant attention due to its broad applications in domains such as finance, traffic, and energy management (Lim & Zohren, 2021; Miller et al., 2024; Sezer et al., 2020; Jiang et al., 2023; Deb et al., 2017). This potential has driven the development of a wide range of approaches, including mathematical, statistical, and deep learning methods.

Recent advances have primarily focused on modeling long-term temporal dependencies and capturing inter-variable relationships. Transformer- and MLP-based models have achieved notable success by incorporating domain-specific properties of time series.

Inspired by progress in natural language processing (NLP) and computer vision (CV), more sophisticated designs have also been applied to TSF. For instance, temporal convolutional network (TCN)-based methods currently frame long-term sequence modeling as the challenge of expanding the effective receptive field (ERF). To address this, extensive efforts have been invested in exploring state-of-the-art (SOTA) techniques to increase network depth and width (Wang et al., 2023; Luo & Wang, 2024; Cheng et al., 2024). Another research direction contrasts channel independence (CI) with channel dependence (CD). CI methods (Nie et al., 2023; Zhou et al., 2023) ignore cross-variable dependencies and predict each variable separately using multiple heads, while CD methods explicitly model inter-variable dependencies and employ a shared prediction head to forecast all variables (Liu et al., 2024; Wu et al., 2023).

Despite this progress, most studies assume that variables share certain statistical consistencies, such as periodicity. Under this assumption, they use a single encoder, like a weakly symmetric function, to jointly model temporal dynamics and inter-variable dependencies, thereby enforcing periodic alignment across variables. However, as shown in Figure 1, when we apply patch-wise attention independently to each variable and visualize the weight relationships between the first patch and the others, the observed periodic patterns contradict this assumption. An alternative strategy is to model temporal dependencies and causal relationships with independent parameters, which we term *asymmetric modeling* to distinguish it from CI. Appendix A.1 provides further analysis of the

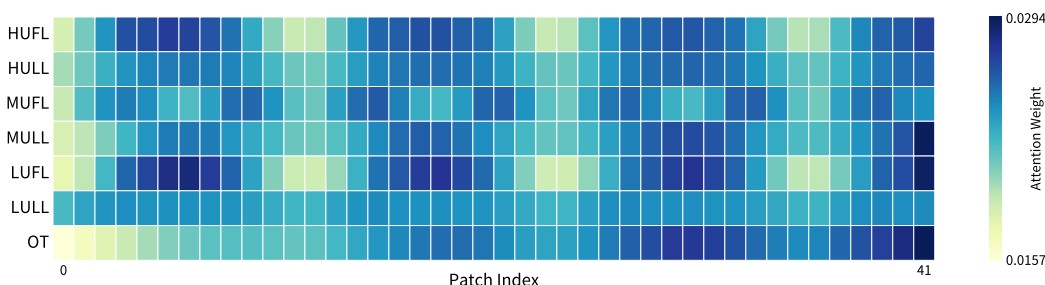

Figure 1: Visualization of patch-wise attention weights on ETTm1. The input sequence of length 336 is divided into 42 patches, with darker colors representing stronger weights. Independent parameter modeling reveals variations in weight magnitudes across rows, reflecting the distinct periodic patterns of each variable.

primary periods in the datasets and confirms that periodicity differs across datasets and variables, underscoring the need for asymmetric modeling.

In summary, the central challenge of multivariate time series modeling is to achieve efficient sequence modeling while maintaining parameter independence. Two natural perspectives are attention mechanisms and convolutional cardinality. Attention has been widely adopted in TSF due to its strong capacity for sequence modeling. In particular, patch-wise attention (Nie et al., 2023), which segments sequences into patches and weights their interactions, has emerged as a robust baseline. However, attention suffers from high computational cost and sensitivity to positional encodings (Chen et al., 2021; Zhou et al., 2022; Wu et al., 2021; Xu et al., 2021). Convolution, on the other hand, incorporates cardinality by dividing channels into groups and applying independent convolutions, while inherently encoding relative positional information. This property complements the limitations of attention (Zhao et al., 2021). Combining the strengths of both paradigms therefore offers a promising direction for advancing sequence modeling in TSF.

Building on these insights, we propose a novel framework centered on the Temporal Convolutional Association Block (TCAB), where the Patch-wise Association Block (PAB) and the Variable-wise Association Block (VAB) represent two variants of its application. TCAB leverages group convolution to realize a patch-wise equivalent yet efficient attention mechanism by replacing attention score computation with learnable weights. This design enables asymmetric modeling of time series. As shown in Figure 2, TCAB independently processes each variable and captures inter-patch temporal dependencies at each time step, thereby modeling asymmetric causal relationships and long-term dependencies across both variables and patches. Based on TCAB, we further develop the Temporal Convolutional Association Network (TCAN), which achieves SOTA performance on seven real-world datasets. Our main contributions are summarized as follows:

- To the best of our knowledge, this is the first work to reveal that existing TSF approaches rely on symmetric modeling, such as sharing a single encoder across variables, which conflicts with the empirical observation that variables in real-world datasets do not share consistent periodicity.

- We propose TCAB, a module that combines the strengths of attention and convolution, retaining the modeling capacity of attention while supporting asymmetric modeling. Building on TCAB, we introduce TCAN, which captures asymmetric temporal and causal relationships effectively.

- We conduct extensive experiments on seven real-world datasets, demonstrating the state-of-the-art performance of TCAN, validating the effectiveness of TCAB, and providing a concrete example of successful asymmetric modeling.

## 2  RELATED WORK

### 2.1  TRANSFORMER-BASED METHODS

In recent years, Transformer-based models have received considerable attention in TSF. We briefly review several representative approaches. Autoformer (Chen et al., 2021) introduces

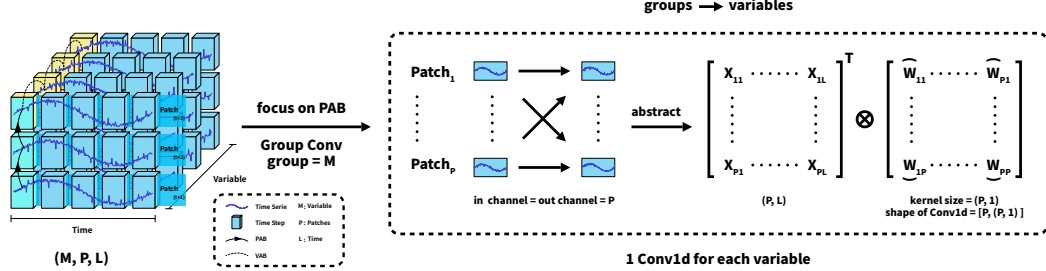

Figure 2: Workflow of TCAB with PAB as an example. Details are provided in the Appendix B.1.

auto-correlation mechanisms and moving averages to enhance temporal pattern modeling. FED-former (Zhou et al., 2022) employs frequency-domain representations with Fourier transforms to achieve linear computational complexity. PatchTST (Nie et al., 2023) applies a patching strategy to strengthen local temporal semantics and reduce attention costs. iTransformer (Liu et al., 2024) inverts the standard Transformer architecture to better capture latent temporal dependencies. SimpleTM (Chen et al., 2025) improves attention performance through tokenization methods inspired by signal processing. Transformer-based models have been widely studied, with efforts directed toward enhancing temporal dependency modeling and reducing the computational cost of attention.

## 2.2 MLP-BASED METHODS

Research on MLPs has also introduced several innovative perspectives for time series modeling. DLinear (Zeng et al., 2023) revisits TSF design with a simple but effective linear decomposition. Koopa (Liu et al., 2023), grounded in Koopman operator theory (Brunton et al., 2021), formulates TSF as a dynamic system identification problem. TimeMixer (Wang et al., 2024) incorporates feature pyramid networks into temporal modeling. FITS (Xu et al., 2024b) applies linear transformations in the complex frequency domain to extract informative temporal features. These MLP-based methods demonstrate that even without explicit recurrence or attention, complex temporal dynamics can be effectively modeled through architectural innovations.

## 2.3 TCN-BASED METHODS

Table 1: Comparison of time series convolutional models.

| Designs | SCINet | TimesNet | MICN | ModernTCN | ConvTimeNet | Ours |
|---|---|---|---|---|---|---|
| Small Kernel | ✓ | ✓ | ✓ | ✗ | ✗ | ✓ |
| Non-Gaussian Receptive Field | ✗ | ✗ | ✗ | ✗ | ✗ | ✓ |
| Asymmetric Modeling | ✗ | ✗ | ✗ | ✗ | ✗ | ✓ |

As convolutional architectures continue to evolve, a resurgence of interest has emerged. Several recent models explore diverse convolutional designs to improve temporal representation learning. SCINet (Liu et al., 2022a) abandons causal convolution and achieves temporal feature fusion through a recursive downsampling–convolution–interaction pipeline. TimesNet (Wu et al., 2023) adapts 2D convolutional backbones from CV to learn expressive temporal representations. MICN (Wang et al., 2023) employs a multi-scale hybrid decomposition module to jointly model local and global temporal dependencies. ModernTCN (Luo & Wang, 2024) designs convolutional architectures combining large kernel convolutions and depthwise separable convolutions (DSC), guided by receptive field analysis. ConvTimeNet (Cheng et al., 2024) stacks large kernel and DSC to enable effective multi-scale temporal modeling.

Unlike most convolutional models that enlarge the ERFs by increasing kernel size or network depth, TCAN achieves sequence modeling equivalent to attention and supports asymmetric modeling while using only kernels of size one, which leads to non-Gaussian receptive fields. As shown in Table 1, other models typically rely on larger kernels to aggregate temporal information through broader Gaussian receptive fields. A proof of the origin of Gaussian receptive fields is provided in Appendix C.1, and a detailed comparative analysis between TCAB and DSC is presented in Appendix B.2.

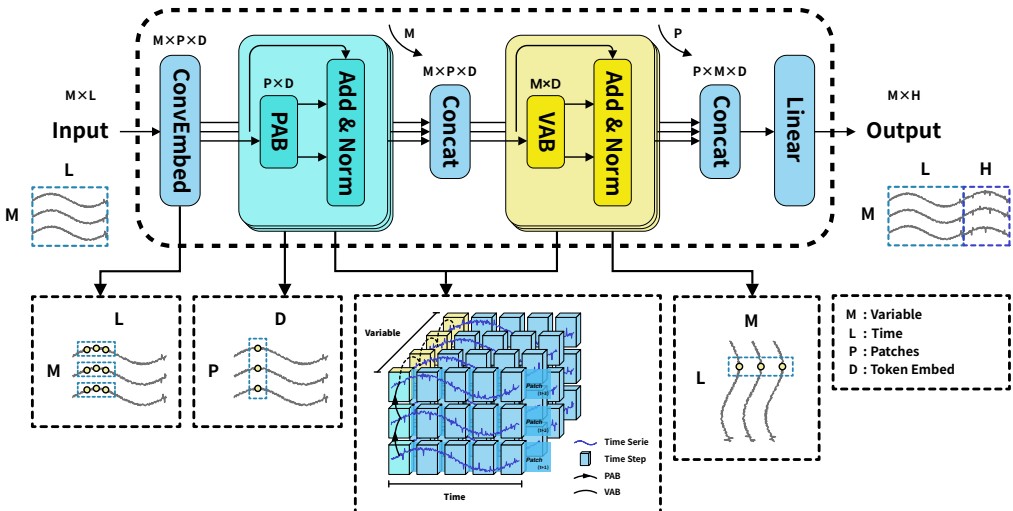

Figure 3: Overall architecture of TCAN consists of ConvEmbed, PAB, and VAB, which address temporal relationships and variable relationships respectively.

## 3 TCAN

In TSF, given historical observations $\mathbf{X} = [\boldsymbol{x}_1, \cdots, \boldsymbol{x}_L] \in \mathbb{R}^{L \times M}$ with $L$ time steps and $M$ variables, we predict the future $H$ time steps $\mathbf{Y} = \{\boldsymbol{x}_{T+1}, \ldots, \boldsymbol{x}_{T+H}\} \in \mathbb{R}^{H \times M}$. In this paper, we propose TCAN, which incorporates two TCABs. Technically, it consists of ConvEmbed block, PAB, and VAB, designed to extract asymmetric temporal and variable relationships, as well as to handle both local features and global dependencies.

### 3.1 Structure Overview

TCAN, shown in Figure 3, adopts a fully convolutional architecture. To mitigate distribution shifts in time series data and enhance the extraction of temporal semantics, we apply instance normalization (IN) and patching, following mainstream studies (Nie et al., 2023; Cheng et al., 2024). ConvEmbed, PAB, and VAB then work together to progressively capture intra-patch temporal features, inter-patch temporal patterns, and inter-variable temporal relationships. The detailed workflows of ConvEmbed, PAB, and VAB are presented in the following subsections.

### 3.2 ConvEmbed

To avoid explicit position encoding, we introduce the ConvEmbed block based on 1D convolution. Let the output $\mathbf{X} \in \mathbb{R}^{M \times P \times D}$ from the Patching layer serve as the input to the ConvEmbed. Here, $M$ represents the number of variables, $P$ denotes the number of patches after Patching, and $D$ is the length of the embedded tokens. The above procedure can be formulated as follows:

$$\text{ConvEmbed}(\mathbf{X}) = \text{GELU}(\text{Conv1D}(\mathbf{X})). \tag{1}$$

Specifically, the kernel size of ConvEmbed is kept consistent with the patch length to ensure information consistency. Then, ConvEmbed is applied within each patch to extract semantic features of adjacent time steps and enhance the semantic representation ability of the model. Furthermore, by sharing semantic extraction patterns across different variables, the model is guided to focus on common semantic features. Finally, the Gaussian Error Linear Unit (GELU) activation function is applied to introduce non-linearity between the blocks.

## 3.3 TCAB

The TCAB module leverages group convolution to combine inter-group information isolation, a bottleneck structure, and temporal invariance for asymmetric temporal dependency modeling. Taking PAB as an example, it assigns each variable to a distinct group equal to the number of variables, with convolutional weights shared only within a variable's temporal patches. This isolates interactions across variables and allows each variable to capture its own temporal patterns, providing a solid foundation for modeling asymmetric long-term dependencies. The workflow of TCAB is given by

$$\text{TCAB}(\mathbf{X}) = \text{Drop}\big(\text{Conv1D}_2(\text{GELU}(\text{Conv1D}_1(\mathbf{X})))\big). \tag{2}$$

**PAB** The ConvEmbed output, reshaped as $\mathbf{X} \in \mathbb{R}^{(M \times P) \times D}$, serves as input to PAB. Designed to capture periodic variations and global representations, PAB processes $\mathbf{X}$ through a grouped one-dimensional convolution:

$$\mathbf{Z}_1 = \text{Conv1D}(\mathbf{X}; \mathbf{W}_1), \tag{3}$$

where $\mathbf{W}_1 \in \mathbb{R}^{M \cdot d_{\text{ff}} \times P \times 1}$ with Group $= M$. This expands to

$$z_1^{(m,p,d)} = \sum_{k=1}^{P} w_1^{(m,p,k)} \cdot x^{(m,k,d)} + b_1^{(m,p)}, \tag{4}$$

where $m \in [1, M]$ denotes the group index, $P \to d_{\text{ff}}$ is the channel expansion factor, and $\mathbf{Z}_1 \in \mathbb{R}^{(M \cdot d_{\text{ff}}) \times D}$. Another convolution then forms a bottleneck structure:

$$\mathbf{Z}_2 = \text{Drop}(\text{Conv1D}(\text{GELU}(\mathbf{Z}_1); \mathbf{W}_2^l)), \tag{5}$$

where $\mathbf{W}_2^l \in \mathbb{R}^{M \cdot P \times d_{\text{ff}} \times 1}$ with Group $= M$. This expands to

$$z_2^{(m,p,d)} = \sum_{k=1}^{d_{\text{ff}}} w_2^{(m,p,k)} \cdot x^{(m,k,d)} + b_2^{(m,p)}, \tag{6}$$

where $d_{\text{ff}} \to P$ is the channel compression factor and $\mathbf{Z}_2 \in \mathbb{R}^{(M \cdot P) \times D}$. This shows that the one-dimensional convolution in PAB essentially acts as a patch association block, computing relationships between local patches within each variable.

**Equivalence to Patch-wise Attention** PAB is mathematically equivalent to patch-wise attention logits under relaxed weight constraints. For patch-wise attention, given $\mathbf{X} \in \mathbb{R}^{M \times P \times D}$ with $\mathbf{Q} = \mathbf{K} = \mathbf{V} = \mathbf{X}$, the attention score between patches $i$ and $j$ of variable $m$ is

$$\mathbf{A}_{i,j}^m = \frac{\exp\left(\langle \mathbf{Q}_i^m, \mathbf{K}_j^m \rangle\right)}{\sum_{j'=1}^{P} \exp\left(\langle \mathbf{Q}_i^m, \mathbf{K}_{j'}^m \rangle\right)}, \tag{7}$$

$$\mathbf{O}_i^m = \sum_{j=1}^{P} \mathbf{A}_{i,j}^m \mathbf{V}_j^m. \tag{8}$$

In PAB, after reshaping $\mathbf{X}$ to $\mathbf{X}' \in \mathbb{R}^{1 \times (M \times P) \times D}$, a grouped one-dimensional convolution is applied. Each group processes $\mathbf{X_m} \in \mathbb{R}^{1 \times P \times D}$ with $(P, 1)$ kernels per channel, yielding $M \times P^2$ learnable weights $\mathbf{W}_{i,j}^m \in \mathbb{R}^D$. The convolution output is

$$\hat{\mathbf{S}}_{i,j}^m = \sum_{d=1}^{D} \mathbf{W}_{i,j,d}^m \mathbf{X}_{j,d}^m. \tag{9}$$

This derivation shows that PAB generates attention logits through learnable weights without query key dot products or softmax normalization. Similar approaches appear in dynamic convolutional attention mechanisms (Wu et al., 2019), which reinterpret attention scores as convolutional weights. By omitting softmax, PAB avoids the constraint of mutual exclusivity, allowing the importance of one patch to increase without diminishing that of others, which is more suitable for TSF.

**VAB** Since VAB and PAB share the same modular design, we describe VAB from a tensor perspective. Given an input $\mathbf{X} \in \mathbb{R}^{M \times P \times D}$, we reshape it to $\mathbf{X}'_{trans} \in \mathbb{R}^{1 \times (P \times M) \times D}$. By setting the number of groups equal to the number of patches, each group convolution processes one patch across variables $\mathbf{X}'_p \in \mathbb{R}^{1 \times M \times D}$. Each output channel corresponds to $(M, 1)$ kernels, operating on an input $\mathbf{X} \in \mathbb{R}^{1 \times M \times D}$. This yields $P \times M^2$ learnable weights that model asymmetric correlations among variables at the same time step, such as the causal asymmetry between temperature and electricity usage.

VAB is therefore obtained by changing the isolation dimension of TCAB from patches to variables. **Together with PAB, it highlights the flexibility of TCAB as a temporal convolution paradigm for multivariate time series.** Its core value lies in balancing information isolation and association within a minimal structure. By using grouped convolutions to decouple data along different dimensions and bottleneck structures for efficient feature transformation, TCAB preserves temporal alignment while enabling asymmetric dependency modeling. To ensure temporal invariance and preserve equivalence to attention, TCAB sets both the kernel size and stride to one within the module. A detailed comparison between TCAB and traditional DSC is provided in Appendix B.2.

### 3.4 INSTANCE NORMALIZATION

This technique, recently proposed to mitigate distribution shift between training and testing data, normalizes each time series instance $x^{(i)}$ to zero mean and unit standard deviation. Specifically, each $x^{(i)}$ is normalized before patching, and the mean and deviation are restored to the output prediction. Mathematically, this process is formulated as:

$$x_{t-L+1:t} = \frac{x_{t-L+1:t} - \mu}{\sqrt{\sigma + \epsilon}}, \tag{10}$$

$$\bar{x}_{t+1:t+H} = \bar{x}_{t+1:t+H} \times \sqrt{\sigma + \epsilon} + \mu, \tag{11}$$

where $\mu$ and $\sigma$ denote the mean and standard deviation of the input window $x_{t-L+1:t}$, respectively, and $\epsilon$ is a small constant for numerical stability. This implementation follows the RevIN approach without learnable affine parameters (Kim et al., 2021).

## 4 EXPERIMENTS

This section evaluates TCAN on a diverse set of TSF tasks, demonstrating its broad applicability and effectiveness. In addition to overall evaluation, we conduct a comprehensive ablation study to quantify the contribution of each individual component within TCAN.

### 4.1 EXPERIMENTAL SETUP

**Datasets** We utilized widely adopted, publicly available real-world benchmark datasets, including Traffic, Electricity, Weather, and four variants of the ETT dataset (ETTh1, ETTh2, ETTm1, ETTm2). Preprocessing procedures, such as dataset segmentation and standardization, follow the protocols used in previous works (Liu et al., 2024; Luo & Wang, 2024).

We carefully selected a set of widely recognized forecasting models as baselines and reran all experiments using their official implementations and provided scripts[1]. The selected baselines include: (1) Transformer-based methods: PatchTST (Nie et al., 2023), iTransformer (Liu et al., 2024), SimpleTM (Chen et al., 2025); (2) Linear-based methods: DLinear (Zeng et al., 2023), FITS (Xu et al., 2024b); (3) Convolution-based methods: TimesNet (Wu et al., 2023), MICN (Wang et al., 2023), ModernTCN (Luo & Wang, 2024), ConvTimeNet (Cheng et al., 2024). Performance was evaluated using Mean Squared Error (MSE) and Mean Absolute Error (MAE).

**Implementation Details** For all datasets, we conducted a hyperparameter search over the lookback window length and evaluated various prediction horizons $H \in \{96, 192, 336, 720\}$. All models were trained using Adam and each experiment was repeated three times to ensure result stability. All models were reproduced using their official implementations and recommended hyperparameters.

---

[1]Following FITS, we also addressed a longstanding bug in the shared training architecture; details can be found in their public codebase.

Table 2: Performance comparison of different models on seven forecasting datasets. Metrics include MSE and MAE for different time horizons. The best results are highlighted in **bold** while the second best are underlined. We provide more detailed results and robustness analysis in Appendix E.

| Method | | | TCN-based | | | | | | | | Transformer-based | | | | | |
|---|---|---|---|---|---|---|---|---|---|---|---|---|---|---|---|---|
| Model | | TCAN (ours) | | ConvTimeNet (2025) | | ModernTCN (2024) | | MICN (2023) | | TimesNet (2023) | | SimpleTM (2025) | | iTransformer (2024) | | PatchTST (2023) |
| Metric | | MSE | MAE | MSE | MAE | MSE | MAE | MSE | MAE | MSE | MAE | MSE | MAE | MSE | MAE | MSE | MAE |
| ETTh1 | 96 | **0.368** | **0.390** | 0.379 | 0.399 | 0.381 | 0.401 | 0.405 | 0.429 | 0.423 | 0.437 | 0.373 | 0.395 | 0.399 | 0.414 | 0.382 | 0.405 |
| | 192 | **0.405** | **0.413** | 0.408 | 0.416 | 0.422 | 0.426 | 0.503 | 0.499 | 0.481 | 0.481 | 0.426 | 0.425 | 0.435 | 0.440 | 0.414 | 0.421 |
| | 336 | **0.424** | **0.427** | 0.438 | 0.436 | 0.442 | 0.440 | 0.476 | 0.482 | 0.489 | 0.478 | 0.469 | 0.450 | 0.457 | 0.456 | 0.431 | 0.435 |
| | 720 | **0.433** | **0.455** | 0.454 | 0.464 | 0.474 | 0.478 | 0.718 | 0.642 | 0.532 | 0.515 | 0.472 | 0.468 | 0.483 | 0.489 | 0.449 | 0.466 |
| ETTh2 | 96 | **0.270** | **0.333** | 0.280 | 0.339 | 0.276 | 0.340 | 0.294 | 0.356 | 0.378 | 0.421 | 0.293 | 0.345 | 0.315 | 0.366 | 0.276 | 0.338 |
| | 192 | **0.334** | **0.378** | 0.342 | 0.381 | 0.343 | 0.388 | 0.415 | 0.446 | 0.409 | 0.439 | 0.379 | 0.398 | 0.388 | 0.409 | 0.339 | 0.379 |
| | 336 | **0.347** | **0.396** | 0.371 | 0.407 | 0.359 | 0.407 | 0.564 | 0.541 | 0.414 | 0.441 | 0.419 | 0.430 | 0.410 | 0.429 | 0.367 | 0.399 |
| | 720 | **0.373** | **0.418** | 0.394 | 0.432 | 0.408 | 0.440 | 1.256 | 0.825 | 0.433 | 0.457 | 0.424 | 0.443 | 0.434 | 0.452 | 0.392 | 0.430 |
| ETTm1 | 96 | **0.286** | **0.342** | 0.292 | 0.344 | 0.302 | 0.353 | 0.305 | 0.354 | 0.344 | 0.378 | 0.324 | 0.364 | 0.303 | 0.356 | 0.293 | 0.343 |
| | 192 | **0.325** | **0.361** | 0.331 | 0.367 | 0.349 | 0.384 | 0.355 | 0.393 | 0.361 | 0.394 | 0.360 | 0.380 | 0.341 | 0.379 | 0.330 | 0.368 |
| | 336 | **0.360** | **0.381** | 0.365 | 0.389 | 0.385 | 0.403 | 0.384 | 0.407 | 0.428 | 0.432 | 0.391 | 0.403 | 0.381 | 0.402 | 0.366 | 0.392 |
| | 720 | **0.417** | **0.415** | 0.433 | 0.423 | 0.440 | 0.437 | 0.445 | 0.442 | 0.462 | 0.456 | 0.454 | 0.437 | 0.443 | 0.438 | 0.420 | 0.425 |
| ETTm2 | 96 | **0.160** | **0.247** | 0.169 | 0.258 | 0.175 | 0.261 | 0.188 | 0.287 | 0.184 | 0.273 | 0.174 | 0.257 | 0.181 | 0.269 | 0.165 | 0.255 |
| | 192 | **0.213** | **0.288** | 0.224 | 0.294 | 0.226 | 0.298 | 0.241 | 0.325 | 0.243 | 0.309 | 0.238 | 0.299 | 0.238 | 0.310 | 0.220 | 0.292 |
| | 336 | **0.266** | **0.322** | 0.279 | 0.330 | 0.277 | 0.331 | 0.372 | 0.386 | 0.303 | 0.350 | 0.294 | 0.336 | 0.292 | 0.344 | 0.277 | 0.329 |
| | 720 | **0.358** | **0.381** | 0.362 | 0.384 | 0.387 | 0.401 | 0.416 | 0.432 | 0.393 | 0.405 | 0.397 | 0.397 | 0.378 | 0.398 | 0.369 | 0.386 |
| Weather | 96 | **0.145** | **0.194** | 0.156 | 0.207 | 0.154 | 0.207 | 0.173 | 0.241 | 0.170 | 0.228 | 0.154 | 0.201 | 0.165 | 0.215 | 0.155 | 0.204 |
| | 192 | **0.188** | **0.238** | 0.198 | 0.245 | 0.201 | 0.252 | 0.217 | 0.264 | 0.215 | 0.264 | 0.206 | 0.249 | 0.211 | 0.256 | 0.195 | 0.241 |
| | 336 | **0.238** | **0.275** | 0.250 | 0.287 | 0.248 | 0.288 | 0.277 | 0.332 | 0.273 | 0.302 | 0.264 | 0.289 | 0.259 | 0.295 | 0.249 | 0.284 |
| | 720 | **0.312** | **0.326** | 0.325 | 0.337 | 0.338 | 0.346 | 0.315 | 0.356 | 0.341 | 0.350 | 0.343 | 0.342 | 0.327 | 0.339 | 0.321 | 0.335 |
| ECL | 96 | **0.130** | 0.228 | 0.132 | 0.227 | 0.135 | 0.231 | 0.150 | 0.261 | 0.176 | 0.283 | 0.146 | 0.240 | 0.131 | 0.227 | 0.131 | **0.223** |
| | 192 | **0.149** | 0.247 | 0.149 | 0.243 | 0.150 | 0.243 | 0.173 | 0.283 | 0.186 | 0.290 | 0.160 | 0.252 | 0.155 | 0.250 | 0.149 | **0.242** |
| | 336 | **0.163** | 0.261 | 0.167 | 0.261 | 0.166 | **0.259** | 0.196 | 0.306 | 0.210 | 0.308 | 0.174 | 0.267 | 0.166 | 0.264 | 0.167 | 0.261 |
| | 720 | **0.189** | **0.286** | 0.206 | 0.293 | 0.208 | 0.298 | 0.302 | 0.386 | 0.226 | 0.321 | 0.208 | 0.296 | 0.222 | 0.318 | 0.202 | 0.292 |
| Traffic | 96 | 0.385 | 0.265 | 0.377 | 0.265 | 0.397 | 0.278 | 0.476 | 0.295 | 0.591 | 0.322 | 0.421 | 0.281 | **0.356** | 0.263 | 0.365 | **0.250** |
| | 192 | 0.398 | 0.270 | 0.396 | 0.272 | 0.415 | 0.287 | 0.488 | 0.304 | 0.609 | 0.328 | 0.442 | 0.290 | **0.369** | 0.269 | 0.383 | **0.258** |
| | 336 | 0.411 | 0.275 | 0.409 | 0.280 | 0.428 | 0.295 | 0.493 | 0.295 | 0.621 | 0.340 | 0.467 | 0.300 | **0.386** | 0.277 | 0.397 | **0.264** |
| | 720 | 0.446 | 0.301 | 0.438 | 0.294 | 0.454 | 0.311 | 0.515 | 0.312 | 0.646 | 0.344 | 0.503 | 0.320 | **0.417** | 0.291 | 0.432 | **0.285** |

## 4.2 MAIN RESULTS

As shown in Table 2, TCAN achieves SOTA performance on most datasets, outperforming MLP-based, Transformer-based, and Convolution-based models. In particular, TCAN surpasses the best-performing TCNs, highlighting the effectiveness of TCAB in TSF. Convolutional models with more complex designs, such as stacked architectures or large kernels, perform poorly on real-world datasets. *This suggests that in TSF, effective convolutional design may be more important than simply enlarging the receptive field.* In contrast, TCAN and PatchTST, both of which adopt patch-wise attention, show competitive performance. The comparison between TCAN and PatchTST further demonstrates the benefit of asymmetric modeling, validating the phenomena observed in Figure 1.

TCAN also achieves SOTA results on ECL, yet it underperforms Transformer-based models on the Traffic dataset, which involves complex spatiotemporal relationships and anomalous events such as delays and dynamic fluctuations (Xu et al., 2024a). To investigate this gap, we examined the distribution of extreme values in Table 3 and found that Traffic contains substantial outliers in both frequency and magnitude. Further analysis of Traffic dataset is provided in Appendix D.

Table 3: Outlier of datasets. the average number and scale of extreme points per window in each dataset when the Z-Score>6 and the window size is 720.

| | Traffic | ECL | Weather | ETTh1 | ETTh2 | ETTm1 | ETTm2 |
|---|---|---|---|---|---|---|---|
| Avg. Count | **610.38** | 22.8 | 3.98 | 0.0 | 0.74 | 0.0 | 0.85 |
| Avg. Scale | **4693.72** | 169.42 | 65.19 | Nan | 4.76 | Nan | 5.38 |

Two factors help explain this observation. First, metric sensitivity plays a role. MSE emphasizes outlier modeling, whereas MAE better reflects general modeling capability. On high-dimensional datasets such as ECL (321 variables) and Traffic (862 variables), TCAN achieves MAE comparable to the second-best model. Second, outlier handling is important. As shown in Appendix E.1, the pattern of MAE being close to the second-best model but MSE showing a larger gap is common

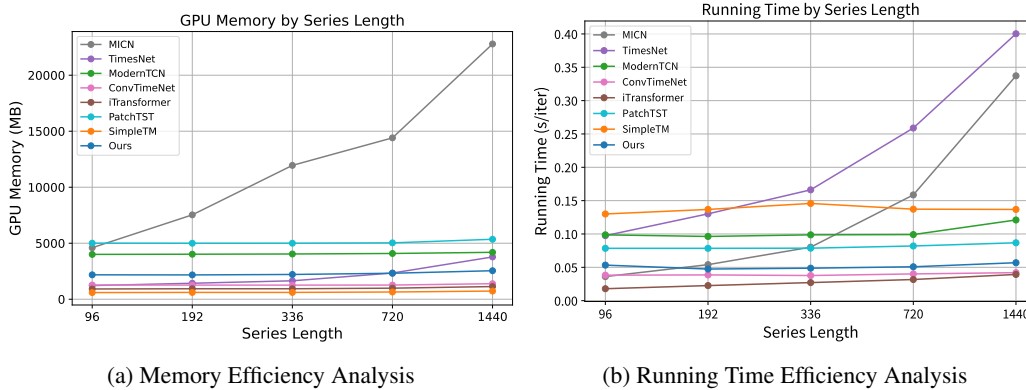

(a) Memory Efficiency Analysis        (b) Running Time Efficiency Analysis

Figure 4: Analysis of memory usage and time efficiency of the model on the Weather dataset. We have further provided results comparing additional models in the Appendix E.

among non-Transformer networks. This suggests that TCAN is less sensitive to outliers than dot-product-based attention methods, which tend to assign disproportionately high weights to extreme values and thereby achieve lower MSE.

To further validate this point, we compared TCAN with PatchTST on the Solar dataset (137 variables, fewer outliers) in Appendix E.2, where TCAN outperformed PatchTST. This confirms that TCAN's weaker performance on Traffic stems mainly from the abundance of outliers rather than from increased dimensionality.

## 4.3 MODEL ANALYSIS

Table 4: Ablation Study on TCAN: We systematically replace or remove components to assess its feature extraction capability. The average results across all predicted lengths are reported. More details can be find in the Appendix E.

| Design | Time | Variable | ETTh2 | | Weather | | Electricity | | Traffic | |
|--------|------|----------|-------|-------|---------|-------|-------------|-------|---------|-------|
| | | | MSE | MAE | MSE | MAE | MSE | MAE | MSE | MAE |
| TCAN | PAB | VAB | **0.331** | **0.381** | **0.221** | **0.258** | **0.158** | **0.255** | **0.410** | **0.278** |
| Replace | MLPFFN | VAB | 0.342 | 0.387 | 0.230 | 0.265 | 0.161 | 0.258 | 0.430 | 0.291 |
| | ConvFFN | VAB | 0.338 | 0.384 | 0.231 | 0.266 | **0.158** | 0.256 | 0.422 | 0.287 |
| w/o | w/o | VAB | 0.340 | 0.386 | 0.232 | 0.267 | 0.160 | 0.257 | 0.428 | 0.290 |
| | PAB | w/o | 0.338 | 0.385 | 0.224 | 0.260 | 0.165 | 0.259 | 0.438 | 0.295 |

**Ablation study** To validate the effectiveness of the TCAN component, we conducted comprehensive ablation studies, which involved both component replacement and removal. The results are presented in Table 4. Notably, the TCAN with TCAB broadly achieves optimal performance. The long-term sequence modeling capability of PAB is particularly influential on low-dimensional datasets such as ETT and Weather. In contrast, for high-dimensional datasets like ECL and Traffic, the ability to model inter-variable relationships becomes increasingly crucial.

Furthermore, Table 4 shows that the independent long-term sequence modeling used in PAB outperforms traditional symmetry modeling such as FFNs, which rely on implicit parameter sharing to capture long-term dependencies. This result further supports the necessity of adopting asymmetric modeling in TSF.

**Efficiency analysis** We compare the running memory and time against the previous SOTA models in Figure 4(a)–(b) under the training phase for various series lengths (ranging from 96 to 1440). It can be observed that TCAN is not sensitive to the input length and exhibits better efficiency compared to TCNs and most Transformers. Notably, despite TCAN utilizing multiple encoders, it

still manages to maintain competitive efficiency. Moreover, compared to PatchTST, which employs equivalent patch-wise associations, TCAN, with its asymmetric modeling strategy, not only delivers superior performance but also maintains better efficiency. However, TCAN is less efficient than iTransformer with respect to model size and training speed, primarily due to iTransformer's omission of attention mechanisms in the temporal dimension, which significantly reduces computational complexity. Overall, considering the accuracy improvement brought by TCAB, TCAN achieves the best balance between performance and efficiency.

**Hyperparameter sensitivity analysis** To see whether TCAN is sensitive to the choice of layer and patch length settings, we perform another experiments with varying model parameters. As Table 5 shown, TCAN is not sensitive to the setting of hyperparameters. Using the unified parameters with PAB = 1, VAB = 3 and patch length = 8 is sufficient to most scenarios.

Table 5: Hyperparameter Sensitivity Analysis.

| Design | Num | ETTm2 | | Weather | | ECL | |
|---|---|---|---|---|---|---|---|
| | | MSE | MAE | MSE | MAE | MSE | MAE |
| PAB | 1 | **0.249** | **0.310** | **0.221** | **0.258** | **0.158** | **0.255** |
| | 2 | 0.252 | 0.312 | 0.222 | 0.258 | 0.16 | 0.258 |
| | 3 | 0.254 | 0.313 | 0.224 | 0.259 | 0.163 | 0.26 |
| | 4 | 0.254 | 0.313 | 0.223 | 0.258 | 0.164 | 0.262 |
| VAB | 1 | 0.253 | 0.313 | 0.222 | 0.26 | 0.164 | 0.262 |
| | 2 | 0.252 | 0.310 | 0.222 | 0.259 | 0.161 | 0.259 |
| | 3 | **0.25** | **0.310** | **0.221** | **0.259** | **0.158** | **0.256** |
| | 4 | 0.252 | 0.312 | 0.224 | 0.261 | 0.158 | 0.255 |
| patch length | 4 | 0.251 | 0.310 | 0.225 | 0.262 | 0.164 | 0.265 |
| | 8 | **0.249** | **0.310** | **0.221** | **0.259** | 0.158 | 0.256 |
| | 16 | 0.253 | 0.312 | 0.224 | 0.26 | **0.158** | **0.255** |
| | 32 | 0.254 | 0.314 | 0.224 | 0.259 | 0.161 | 0.258 |

## 5 DISCUSSION

**Potential limitations** While TCAN demonstrates strong performance in TSF, it presents several potential limitations that warrant further discussion:

- **Cost of Asymmetric Modeling:** Although TCAN may be more cost-effective than most TCNs and Transformers on many datasets, it incurs additional parameter overhead on high-dimensional datasets such as Traffic, where the parameter size scales with the number of variables due to asymmetric modeling.

- **Impact of outliers:** When a dataset contains significant outliers, the performance of TCAN may be affected. Because TCAB relies on learnable patch weights, it is less responsive to extreme values than inner product-based attention mechanisms, which tend to assign disproportionately high weights to anomalies. This limitation can reduce prediction accuracy in highly irregular or noisy settings.

**Interesting finding** However, as demonstrated in Appendix C.1, TCAN provides a SOTA solution by leveraging non-Gaussian receptive fields. This highlights the significant potential of designing domain-adaptive convolutional structures for TSF. Specifically, the domain-specific designs in TCAN, including asymmetric modeling of temporal and causal relationships and equivalent attention convolution, suggest that tailoring convolutional architectures to the unique characteristics of TSF is a promising direction for future research. In this context, as advanced research shifts toward time series domains, it may become increasingly important to focus on the specific characteristics of temporal data.

## 6 CONCLUSION

In this paper, we reveal that existing approaches in time series forecasting (TSF) typically rely on symmetric modeling, which fails to capture the distinct periodic behaviors observed in real-world datasets. To address this limitation, we propose the Temporal Convolutional Association Block (TCAB), a flexible module that integrates the strengths of both attention and convolution to support asymmetric modeling across temporal or variable dimensions. Building upon TCAB, we introduce the Temporal Convolutional Association Network (TCAN), which effectively captures asymmetric temporal and causal relationships. Our experimental results affirm the potential of asymmetric modeling as a promising research direction for TSF and highlight TCAB as a principled and efficient approach for advancing multivariate time series forecasting.

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

# A PERIODICITY ANALYSIS OF VARIABLES IN EACH DATASET

## A.1 PATCH-WISE ATTENTION ON THE ETT DATASETS

To examine the necessity of asymmetric modeling, we apply patch-wise attention on the ETT datasets and visualize the relationships between the first patch and subsequent patches across different variables. The visualization highlights variable-specific periodic structures, demonstrating that periodicity is not consistent across variables.

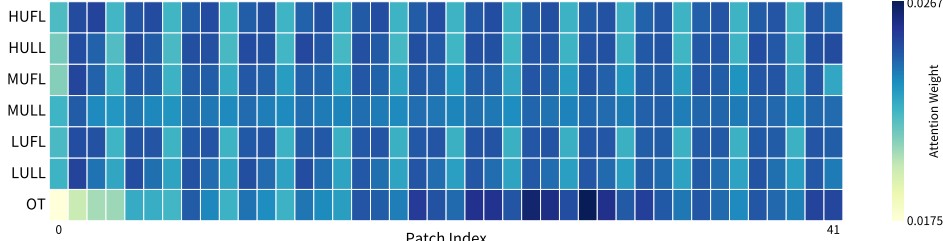

Figure 5: Visualization of Patch-wise Attention on ETTh1

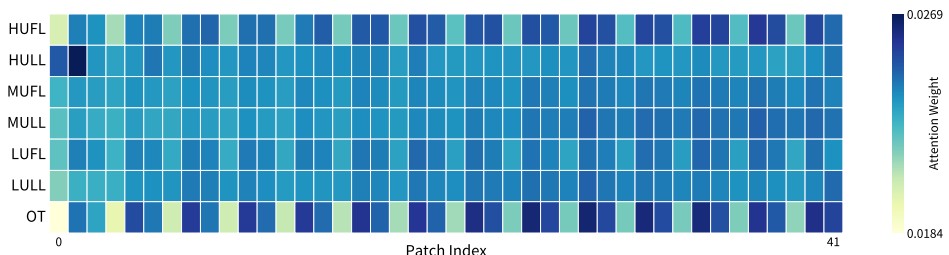

Figure 6: Visualization of Patch-wise Attention on ETTh2

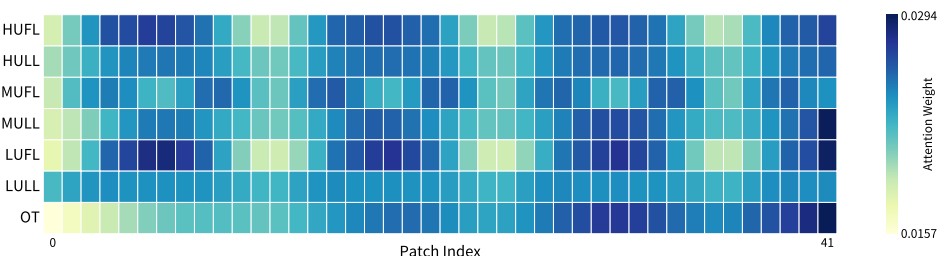

Figure 7: Visualization of Patch-wise Attention on ETTm1.

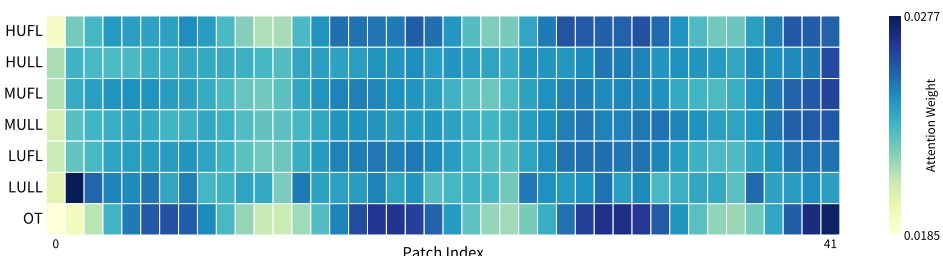

Figure 8: Visualization of Patch-wise Attention on ETTm2.

## A.2 PRINCIPAL PERIOD ANALYSIS BASED ON ACF

The periodicity of variables can be analyzed from two complementary perspectives. At the dataset level, distinct datasets exhibit different periodic behaviors, as widely reported in prior studies such as Weather and ECL (Xu et al., 2024a). At the variable level, even within a single dataset, variables may display heterogeneous periodicity. As shown in Appendix A.1, the results reveal clear periodic patterns, with variables such as HUFL and MUFL exhibiting distinct periods.

To further support this observation, we extend the autocorrelation function (ACF) method from (Xu et al., 2024a). This method, originally applied at the dataset level, is adapted here to analyze periodicity within individual variables. The ACF quantifies autocorrelation by measuring the correlation between a sequence and its lagged values, defined as

$$ACF = \frac{\sum_{t=1}^{N-k}(x_t - \bar{x})(x_{t+k} - \bar{x})}{\sum_{t=1}^{N}(x_t - \bar{x})^2}, \tag{12}$$

where $N$ denotes the number of observations, $x_t$ represents the value at time $t$, $k$ is the lag, and $\bar{x}$ is the mean. Significant peaks in the ACF curve indicate periodicity at the corresponding lag.

Our empirical analysis in Table 6 confirms that periodicity differs not only across variables within the same dataset but also for the same variable across different datasets. For instance, ETTh1 and ETTh2 are recorded at hourly intervals, while ETTm1 and ETTm2 are recorded at 15-minute intervals, leading to variations in their periodic patterns.

Table 6: Periodicity analysis of variables across datasets

| Variable | Major Period | | | | All Periods | | | |
|---|---|---|---|---|---|---|---|---|
| | ETTh1 | ETTh2 | ETTm1 | ETTm2 | ETTh1 | ETTh2 | ETTm1 | ETTm2 |
| HUFL | 24 | 24 | 96 | 58 | 24 | 24 | 96 | 58, 96 |
| HULL | 15 | 15 | 96 | 46 | 15, 24, 39 | 15, 24, 39 | 96 | 46, 58, 87 |
| MUFL | 24 | 24 | 96 | 96 | 24 | 24 | 96 | 96, 142 |
| MULL | 12 | 15 | 59 | 46 | 12, 15, 24 | 15, 24, 36 | 59, 96, 132 | 46, 59, 66 |
| LUFL | 11 | 24 | 45 | 59 | 11, 13, 24 | 24, 47 | 45, 96, 141 | 59, 94, 157 |
| LULL | 17 | - | 68 | - | 17, 24 | - | 68, 96 | - |
| OT | 22 | 24 | 88 | 95 | 22, 48 | 24 | 88, 188 | 95, 191 |

# B DETAILS OF TCAB

## B.1 VISUALIZATION OF TCAB

Figure 9 presents a visualization of the two variants of TCAB, namely PAB and VAB.

## B.2 COMPARISON BETWEEN DSC AND TCAB

Depthwise Separable Convolution (DSC) and the Temporal Convolutional Association Block (TCAB) adopt fundamentally different strategies for information interaction and grouping. A detailed comparison between Depthwise Convolution (DWConv) and TCAB, using PAB as an example, is illustrated in Figure 10.

The N-dimensional DWConv is typically derived from downsampling the D-dimensional input at multiple granularities (Luo & Wang, 2024). DSC employs a combination of DWConv and Pointwise Convolution to decouple spatial and channel information, aiming to enhance expressiveness while mitigating the parameter growth associated with traditional convolutions.

In contrast, PAB within TCAB isolates variables while simultaneously capturing spatiotemporal dependencies in a unified module. This design preserves independent group learning, facilitates patchwise information association within groups, and produces non-Gaussian receptive fields. By combining these properties, PAB introduces a novel convolutional association block for TSF, demonstrating that effective modeling can be achieved through a minimal yet efficient structure.

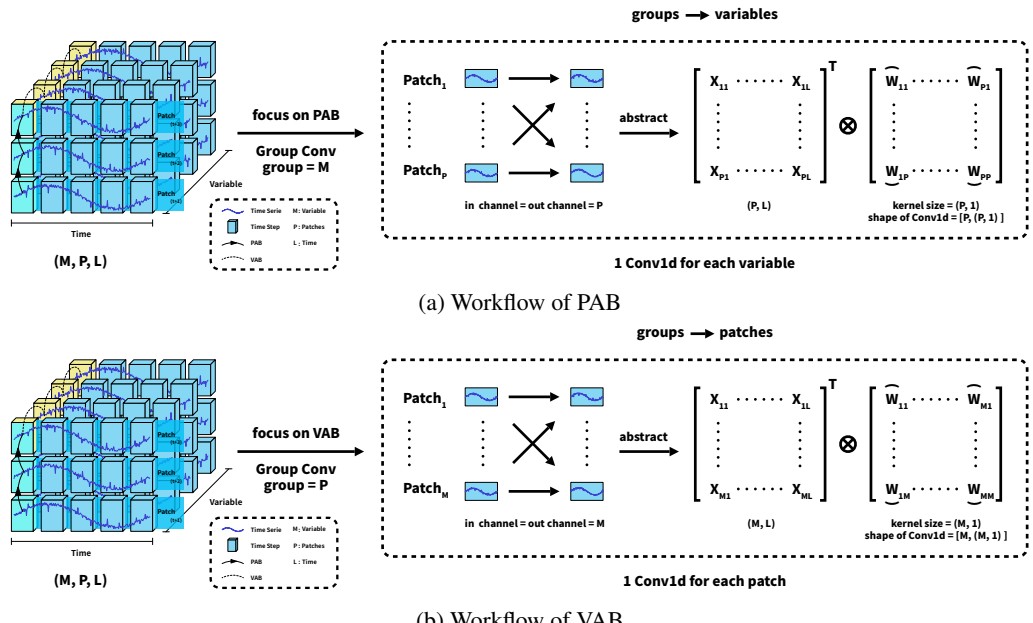

(a) Workflow of PAB

(b) Workflow of VAB

Figure 9: Visualization of the two variants of TCAB.

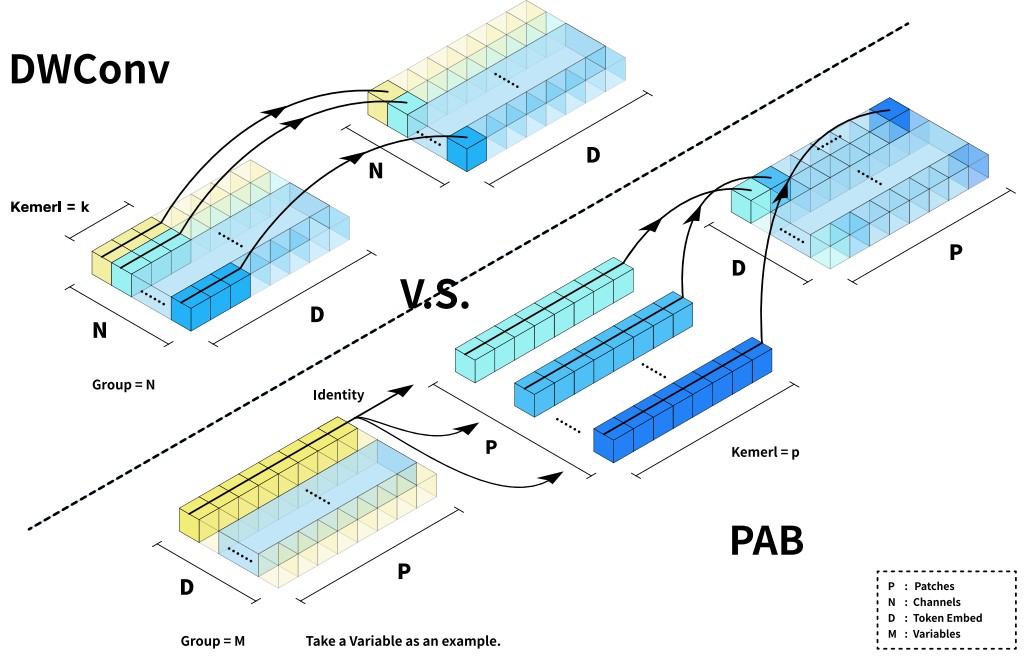

Figure 10: Comparison between DWConv and TCAB.

## C PROOF

### C.1 A SIMPLE PROOF OF THE GAUSSIAN RECEPTIVE FIELD

We consider a convolutional layer with all weights equal to one to provide a simple proof, while a more detailed derivation can be found in Paper (Luo et al., 2017). Assume a stack of $n$ convolutional layers, each using $k \times k$ kernels with stride one, a single channel per layer, and no nonlinearity, forming a deep linear CNN.

Let $g(i,j,p) = \frac{\partial l}{\partial x_{i,j}^p}$ denote the gradient on the $p$-th layer, and let $g(i,j,n) = \frac{\partial l}{\partial y_{i,j}}$. Then $g(,,0)$ corresponds to the gradient image of the input. The backpropagation process convolving $g(,,p)$ with the $k \times k$ kernel produces $g(,,p-1)$ for each $p$.

Since the kernel is a $k \times k$ matrix of ones, the 2D convolution decomposes into two 1D convolutions. We therefore focus on the 1D case. The initial gradient signal $u(t)$ and kernel $v(t)$ are defined as

$$u(t) = \delta(t), \tag{13}$$

$$v(t) = \sum_{m=0}^{k-1} \delta(t-m), \tag{14}$$

where $\delta(t) = \begin{cases} 1, & t = 0 \\ 0, & t \neq 0 \end{cases}$ and $t \in \mathbb{Z}$ indexes the pixels.

The gradient signal on the input pixels is $o = u * v * \cdots * v$, convolving $u$ with $n$ such kernels. To compute this convolution, we apply the Discrete Time Fourier Transform:

$$U(\omega) = \sum_{t=-\infty}^{\infty} u(t)e^{-j\omega t} = 1, \tag{15}$$

$$V(\omega) = \sum_{t=-\infty}^{\infty} v(t)e^{-j\omega t} = \sum_{m=0}^{k-1} e^{-j\omega m}. \tag{16}$$

By the convolution theorem, the Fourier transform of $o$ is

$$\mathcal{F}(o)(\omega) = U(\omega) \cdot V(\omega)^n = \left( \sum_{m=0}^{k-1} e^{-j\omega m} \right)^n. \tag{17}$$

Applying the inverse Fourier transform yields

$$o(t) = \frac{1}{2\pi} \int_{-\pi}^{\pi} \left( \sum_{m=0}^{k-1} e^{-j\omega m} \right)^n e^{j\omega t} d\omega, \tag{18}$$

$$\frac{1}{2\pi} \int_{-\pi}^{\pi} e^{-j\omega s} e^{j\omega t} d\omega = \begin{cases} 1, & s = t \\ 0, & s \neq t \end{cases}. \tag{19}$$

Thus $o(t)$ corresponds to the coefficient of $e^{-j\omega t}$ in the expansion of $\left( \sum_{m=0}^{k-1} e^{-j\omega m} \right)^n$.

## C.2 THE ELABORATION OF NON-GAUSSIAN RECEPTIVE FIELD

This section provides an explanation of why the receptive field becomes discrete when the kernel size equals the stride, as in the case of TCAB.

### C.2.1 NECESSITY

In Section C.1, the Gaussian receptive field derivation assumes stride equal to one, kernel size equal to $k$, and weights fixed at one for all convolution layers. The initial gradient signal and kernel are represented as $u(t) = \delta(t)$ and $v(t) = \sum_{m=0}^{k-1} \delta(t-m)$, with the input gradient given by $u * v^n$. Under stride one, the convolution theorem transforms the gradient into a frequency-domain product. By the Central Limit Theorem, the inverse transform coefficients approximate a Gaussian distribution due to multi-path superposition. When stride equals kernel size, however, the output at the $p$-th layer $x_i^p$ depends only on the discrete block $[i \cdot k, i \cdot k + k - 1]$ from the $(p-1)$-th layer, eliminating continuous overlap. In this setting, the backpropagation gradient becomes

$$g(i, p-1) = \sum_{m=0}^{k-1} w_m \cdot g(i \cdot k + m, p),$$

rather than

$$g(i, p - 1) = \sum_{m=0}^{k-1} w_m \cdot g(i + m, p).$$

Because the convolution theorem requires continuous sliding windows, the Fourier-based Gaussian derivation does not apply when stride equals kernel size.

### C.2.2 Sufficiency

When stride equals kernel size, the output gradient influences the input only through discrete and non-overlapping blocks. This can be written as

$$g(j, p - 1) = \sum_i g(i, p) \cdot w[j - i \cdot k],$$

which is nonzero only when $j \in [i \cdot k, i \cdot k + k - 1]$. Since there are no gradients connecting adjacent blocks, the resulting distribution is discrete and block-like, lacking the smooth continuity and decay that characterize a Gaussian distribution.

## D Experiment Details

### D.1 Datasets

We evaluate the performance of our method on seven real-world IoT datasets. The ETT (Electricity Transformer Temperature) dataset contains two years of data collected from two counties in China, with subsets designed for different granularities of forecasting. ETTh1 and ETTh2 are recorded hourly, while ETTm1 and ETTm2 are recorded every 15 minutes. The ECL dataset records the hourly electricity consumption of 321 customers. The Traffic dataset includes 862 measurements such as vehicle counts, speed, and congestion levels collected by sensors and cameras across the San Francisco Bay area from 2015 to 2016. The Weather dataset consists of 21 meteorological variables including temperature, precipitation, wind speed, and humidity, recorded every 10 minutes throughout 2020 in Germany.

Table 7: The detail statistics of datasets

| Datasets Name | Timesteps | Frequency | Variable |
|---|---|---|---|
| Weather | 52696 | 10 min | 21 |
| Electricity | 26304 | 1 hour | 321 |
| Traffic | 17544 | 1 hour | 862 |
| Exchange | 7207 | 1 day | 8 |
| ETTh1 | 17420 | 1 hour | 7 |
| ETTh2 | 17420 | 1 hour | 7 |
| ETTm1 | 69680 | 15 min | 7 |
| ETTm2 | 69680 | 15 min | 7 |

Table 7 provides detailed statistics of these datasets. *Timesteps* denotes the total number of observations, *Frequency* represents the sampling interval, and *Variables* indicates the number of recorded features.

### D.2 Analysis of Datasets

**Non-stationary Analysis** We apply the Augmented Dick-Fuller (ADF) test statistic (Elliott et al., 1992; Liu et al., 2022b) to measure the degree of stationarity. A smaller ADF statistic reflects stronger stationarity, indicating that the distribution is more stable. Table 7 summarizes the overall statistics of the datasets, presented in ascending order by stationarity level.

We adopt the Augmented Dick-Fuller (ADF) test statistic(Elliott et al., 1992; Liu et al., 2022b) as the metric to quantitatively measure the *degree of stationarity*. A smaller ADF test statistic indicates a higher degree of stationarity, which means the distribution is more stable. Table 1 summarizes the overall statistics of the datasets.

**Outlier Analysis** We also investigate the statistical characteristics of the datasets (Xu et al., 2024a) to examine the role of outliers. Our analysis reveals that the Traffic dataset contains a particularly large number of extreme values, both in frequency and magnitude, which highlights its challenging nature for forecasting tasks.

Table 8: ADF of datasets. Smaller ADF test statistic indicates more stationary dataset.

| | Traffic | ECL | Weather | ETT (4 subsets) |
|---|---|---|---|---|
| ADF | -15.02 | -8.44 | **-26.68** | -7.67 |

Table 9: Outlier of datasets. the average number and scale of extreme points per window in each dataset when the Z-Score>6 and the window size is 720.

| | Traffic | ECL | Weather | ETTh1 | ETTh2 | ETTm1 | ETTm2 |
|---|---|---|---|---|---|---|---|
| Avg. Count | **610.38** | 22.8 | 3.98 | 0.0 | 0.74 | 0.0 | 0.85 |
| Avg. Scale | **4693.72** | 169.42 | 65.19 | nan | 4.76 | nan | 5.38 |

### D.3 ENVIRONMENTS

All experiments were implemented in PyTorch and executed on a single NVIDIA GeForce RTX 4090 GPU with 24 GB of memory.

## E MORE EXPERIMENTAL RESULTS

### E.1 FULL MAIN RESULTS

Due to space constraints in the main text, we have included all experimental results in Table 10, along with comparisons against MLP-based methods such as FITS, and DLinear. The experimental results further demonstrate the effectiveness of the proposed method.

Table 10: Performance comparison of different models on seven forecasting datasets. Metrics include MSE and MAE for different time horizons. The random seed is fixed as 2021 and the best results are highlighted in **bold** while the second best are underlined.

| | | TCN-based | | | | | | | | | | Transformer-based | | | | | | MLP-based | | | |
|---|---|---|---|---|---|---|---|---|---|---|---|---|---|---|---|---|---|---|---|---|---|
| | | TCAN (ours) | | ConvTimeNet (2025) | | ModernTCN (2024) | | MICN (2023) | | TimesNet (2023) | | SimpleTM (2025) | | iTransformer (2024) | | PatchTST (2023) | | FITS (2024) | | DLinear (2022) | |
| | | MSE | MAE | MSE | MAE | MSE | MAE | MSE | MAE | MSE | MAE | MSE | MAE | MSE | MAE | MSE | MAE | MSE | MAE | MSE | MAE |
| ETTh1 | 96 | **0.368** | **0.390** | 0.379 | 0.399 | 0.381 | 0.401 | 0.405 | 0.429 | 0.423 | 0.437 | 0.373 | 0.395 | 0.399 | 0.414 | 0.382 | 0.405 | 0.374 | 0.395 | 0.384 | 0.405 |
| | 192 | **0.405** | **0.413** | 0.408 | 0.416 | 0.422 | 0.426 | 0.503 | 0.499 | 0.481 | 0.481 | 0.426 | 0.425 | 0.435 | 0.440 | 0.414 | 0.421 | 0.407 | 0.414 | 0.444 | 0.450 |
| | 336 | **0.424** | **0.427** | 0.438 | 0.436 | 0.442 | 0.440 | 0.476 | 0.482 | 0.469 | 0.450 | 0.457 | 0.456 | 0.431 | 0.435 | 0.429 | 0.428 | | | 0.447 | 0.448 |
| | 720 | 0.433 | 0.455 | 0.454 | 0.464 | 0.474 | 0.478 | 0.718 | 0.642 | 0.532 | 0.515 | 0.472 | 0.468 | 0.483 | 0.489 | 0.449 | 0.466 | **0.425** | **0.446** | 0.504 | 0.515 |
| ETTh2 | 96 | **0.270** | **0.333** | 0.280 | 0.339 | 0.276 | 0.340 | 0.294 | 0.356 | 0.378 | 0.421 | 0.293 | 0.345 | 0.315 | 0.366 | 0.276 | 0.338 | 0.274 | 0.337 | 0.290 | 0.353 |
| | 192 | **0.334** | **0.378** | 0.342 | 0.381 | 0.343 | 0.388 | 0.415 | 0.446 | 0.409 | 0.439 | 0.379 | 0.398 | 0.388 | 0.409 | 0.339 | 0.379 | 0.337 | 0.378 | 0.389 | 0.422 |
| | 336 | **0.347** | **0.396** | 0.371 | 0.407 | 0.359 | 0.407 | 0.564 | 0.541 | 0.414 | 0.441 | 0.419 | 0.430 | 0.410 | 0.429 | 0.367 | 0.399 | 0.360 | 0.398 | 0.463 | 0.473 |
| | 720 | **0.373** | **0.418** | 0.394 | 0.432 | 0.408 | 0.440 | 1.256 | 0.825 | 0.433 | 0.457 | 0.424 | 0.443 | 0.434 | 0.452 | 0.392 | 0.430 | 0.386 | 0.423 | 0.733 | 0.606 |
| ETTm1 | 96 | **0.286** | **0.342** | 0.292 | 0.344 | 0.302 | 0.353 | 0.305 | 0.354 | 0.344 | 0.378 | 0.324 | 0.364 | 0.303 | 0.356 | 0.293 | 0.343 | 0.303 | 0.345 | 0.301 | 0.345 |
| | 192 | **0.325** | **0.361** | 0.331 | 0.367 | 0.349 | 0.384 | 0.355 | 0.393 | 0.361 | 0.394 | 0.360 | 0.380 | 0.341 | 0.379 | 0.330 | 0.368 | 0.337 | 0.365 | 0.336 | 0.366 |
| | 336 | **0.360** | **0.381** | 0.365 | 0.389 | 0.385 | 0.403 | 0.384 | 0.407 | 0.428 | 0.432 | 0.391 | 0.403 | 0.381 | 0.402 | 0.366 | 0.392 | 0.372 | 0.385 | 0.372 | 0.389 |
| | 720 | **0.417** | **0.415** | 0.433 | 0.423 | 0.440 | 0.437 | 0.445 | 0.442 | 0.462 | 0.456 | 0.454 | 0.437 | 0.443 | 0.438 | 0.420 | 0.425 | 0.428 | 0.416 | 0.427 | 0.423 |
| ETTm2 | 96 | **0.160** | **0.247** | 0.169 | 0.258 | 0.175 | 0.261 | 0.188 | 0.287 | 0.184 | 0.273 | 0.174 | 0.257 | 0.181 | 0.269 | 0.165 | 0.255 | 0.165 | 0.255 | 0.172 | 0.267 |
| | 192 | **0.213** | **0.288** | 0.224 | 0.294 | 0.226 | 0.298 | 0.241 | 0.325 | 0.243 | 0.309 | 0.238 | 0.299 | 0.238 | 0.310 | 0.220 | 0.292 | 0.220 | 0.291 | 0.238 | 0.314 |
| | 336 | **0.266** | **0.322** | 0.279 | 0.330 | 0.277 | 0.331 | 0.372 | 0.386 | 0.303 | 0.350 | 0.294 | 0.336 | 0.292 | 0.344 | 0.277 | 0.329 | 0.274 | 0.326 | 0.295 | 0.359 |
| | 720 | **0.358** | **0.381** | 0.362 | 0.384 | 0.387 | 0.401 | 0.416 | 0.432 | 0.393 | 0.405 | 0.397 | 0.397 | 0.378 | 0.386 | 0.367 | 0.383 | 0.367 | 0.383 | 0.447 | 0.439 |
| Weather | 96 | **0.145** | **0.194** | 0.156 | 0.207 | 0.154 | 0.207 | 0.173 | 0.241 | 0.170 | 0.228 | 0.154 | 0.201 | 0.165 | 0.215 | 0.155 | 0.204 | 0.145 | 0.196 | 0.174 | 0.233 |
| | 192 | **0.188** | **0.238** | 0.198 | 0.245 | 0.201 | 0.252 | 0.217 | 0.283 | 0.215 | 0.264 | 0.206 | 0.249 | 0.211 | 0.256 | 0.195 | 0.241 | 0.189 | 0.238 | 0.218 | 0.278 |
| | 336 | **0.238** | **0.275** | 0.250 | 0.287 | 0.248 | 0.288 | 0.277 | 0.332 | 0.273 | 0.302 | 0.264 | 0.289 | 0.259 | 0.295 | 0.249 | 0.284 | 0.241 | 0.278 | 0.263 | 0.314 |
| | 720 | **0.312** | **0.326** | 0.325 | 0.337 | 0.338 | 0.346 | 0.315 | 0.356 | 0.341 | 0.350 | 0.343 | 0.342 | 0.327 | 0.339 | 0.321 | 0.335 | 0.319 | 0.333 | 0.332 | 0.374 |
| ECL | 96 | **0.130** | 0.228 | 0.132 | 0.227 | 0.135 | 0.231 | 0.150 | 0.261 | 0.176 | 0.283 | 0.146 | 0.240 | 0.131 | 0.227 | 0.131 | **0.223** | 0.141 | 0.237 | | |
| | 192 | **0.149** | 0.247 | 0.149 | 0.243 | 0.150 | 0.243 | 0.173 | 0.283 | 0.186 | 0.290 | 0.160 | 0.252 | 0.155 | 0.250 | 0.149 | **0.242** | 0.155 | 0.249 | 0.154 | 0.251 |
| | 336 | **0.163** | 0.261 | 0.167 | 0.261 | 0.166 | **0.259** | 0.196 | 0.306 | 0.210 | 0.308 | 0.174 | 0.267 | 0.166 | 0.264 | 0.167 | 0.261 | 0.172 | 0.265 | 0.169 | 0.268 |
| | 720 | **0.189** | **0.286** | 0.206 | 0.293 | 0.208 | 0.298 | 0.302 | 0.386 | 0.226 | 0.321 | 0.208 | 0.296 | 0.222 | 0.318 | 0.202 | 0.292 | 0.210 | 0.297 | 0.204 | 0.301 |
| Traffic | 96 | 0.385 | 0.265 | 0.377 | 0.265 | 0.397 | 0.278 | 0.476 | 0.295 | 0.591 | 0.322 | **0.356** | 0.263 | | | 0.365 | **0.250** | 0.411 | 0.280 | 0.413 | 0.287 |
| | 192 | 0.398 | 0.265 | 0.396 | 0.272 | 0.415 | 0.287 | 0.488 | 0.304 | 0.609 | 0.333 | 0.442 | 0.290 | **0.369** | 0.269 | 0.383 | **0.258** | 0.424 | 0.284 | 0.424 | 0.290 |
| | 336 | 0.411 | 0.275 | 0.409 | 0.280 | 0.428 | 0.295 | 0.493 | 0.295 | 0.621 | 0.340 | 0.467 | 0.300 | **0.386** | 0.277 | 0.397 | **0.264** | 0.436 | 0.290 | 0.438 | 0.299 |
| | 720 | 0.446 | 0.301 | 0.438 | 0.294 | 0.454 | 0.311 | 0.515 | 0.312 | 0.646 | 0.344 | 0.503 | 0.320 | **0.417** | 0.291 | 0.432 | **0.285** | 0.464 | 0.307 | 0.466 | 0.316 |

### E.2 RESULTS ON THE SOLAR DATASET

To further investigate the impact of dataset characteristics on model performance, we evaluate TCAN and PatchTST on the high-dimensional Solar dataset, which contains 137 variables but fewer outliers compared with Traffic. This experiment is designed to separate the influence of dimensionality from that of outliers.

As shown in Table 11, TCAN consistently outperforms PatchTST across all prediction horizons in both MSE and MAE. The margins are particularly clear for shorter horizons such as 96 and 192, where TCAN achieves lower error values. These results confirm that the weaker performance of TCAN on the Traffic dataset is largely attributable to the abundance of outliers rather than to increased dimensionality. The Solar dataset thus provides additional evidence that TCAN maintains robust performance in high-dimensional but relatively clean environments.

Table 11: Performance comparison on the high-dimensional Solar dataset.

| Horizon | TCAN | | PatchTST | |
|---|---|---|---|---|
| | MSE | MAE | MSE | MAE |
| 96 | **0.175** | **0.230** | 0.199 | 0.259 |
| 192 | **0.193** | **0.242** | 0.210 | 0.263 |
| 336 | **0.204** | **0.254** | 0.206 | 0.284 |
| 720 | **0.215** | **0.253** | 0.216 | 0.270 |

### E.3 ROBUSTNESS ANALYSIS

We conducted experiments across seven datasets using random seeds from 2020, 2021, and 2022. The results show standard deviations below 0.001 in most cases, indicating strong model robustness.

### E.4 FULL ABLATION RESULTS

Due to the limited pages, we list the overall ablation study results on the effect of PAB and VAB in TCAN as shown in Table 13. The detailed ablations contain two type of experiments denoted as removing components (w/o) and replacing components (replace).

Table 12: Robustness experiments on different datasets.

| Dataset | MSE | STD | MAE | STD |
|---|---|---|---|---|
| ETTh1 | 0.408 | 0.0005 | 0.422 | 0.0004 |
| ETTh2 | 0.333 | 0.0007 | 0.383 | 0.0005 |
| ETTm1 | 0.348 | 0.0006 | 0.375 | 0.0003 |
| ETTm2 | 0.251 | 0.0012 | 0.310 | 0.0006 |
| Weather | 0.221 | 0.0004 | 0.258 | 0.0003 |
| ECL | 0.158 | 0.0005 | 0.256 | 0.0006 |
| Traffic | 0.412 | 0.0013 | 0.279 | 0.0008 |

### E.5 FULL EFFICIENCY ANALYSIS

As shown in Figure 11, we further compare the efficiency of our model with that of MLPs.

Table 13: Ablation Results on Key Components of TCAN: Impact of PAB, and VAB on Time and Variable Dimensions

| Design | Time | Variable | Prediction Lengths | ETTh2 MSE | ETTh2 MAE | Weather MSE | Weather MAE | Electricity MSE | Electricity MAE | Traffic MSE | Traffic MAE |
|---|---|---|---|---|---|---|---|---|---|---|---|
| TCAN | PAB | VAB | 96 | 0.270 | 0.333 | 0.145 | 0.194 | 0.130 | 0.228 | 0.385 | 0.265 |
| | | | 192 | 0.334 | 0.378 | 0.188 | 0.238 | 0.149 | 0.247 | 0.398 | 0.270 |
| | | | 336 | 0.347 | 0.396 | 0.238 | 0.275 | 0.163 | 0.261 | 0.411 | 0.275 |
| | | | 720 | 0.373 | 0.418 | 0.312 | 0.326 | 0.189 | 0.284 | 0.446 | 0.301 |
| | | | Avg | **0.331** | **0.381** | **0.221** | **0.258** | **0.158** | **0.255** | **0.410** | **0.278** |
| Replace | MLPFFN | VAB | 96 | 0.271 | 0.333 | 0.153 | 0.202 | 0.132 | 0.232 | 0.406 | 0.280 |
| | | | 192 | 0.335 | 0.379 | 0.202 | 0.247 | 0.149 | 0.247 | 0.422 | 0.287 |
| | | | 336 | 0.363 | 0.404 | 0.248 | 0.282 | 0.163 | 0.261 | 0.429 | 0.288 |
| | | | 720 | 0.399 | 0.432 | 0.318 | 0.330 | 0.199 | 0.290 | 0.461 | 0.310 |
| | | | Avg | 0.342 | 0.387 | 0.230 | 0.265 | 0.161 | 0.258 | 0.430 | 0.291 |
| | ConvFFN | VAB | 96 | 0.270 | 0.333 | 0.148 | 0.199 | 0.132 | 0.232 | 0.398 | 0.275 |
| | | | 192 | 0.334 | 0.377 | 0.205 | 0.250 | 0.150 | 0.248 | 0.408 | 0.279 |
| | | | 336 | 0.361 | 0.402 | 0.250 | 0.283 | 0.159 | 0.259 | 0.426 | 0.289 |
| | | | 720 | 0.388 | 0.425 | 0.323 | 0.331 | 0.190 | 0.284 | 0.454 | 0.304 |
| | | | Avg | 0.338 | 0.384 | 0.231 | 0.266 | **0.158** | 0.256 | 0.422 | 0.287 |
| w/o | w/o | VAB | 96 | 0.271 | 0.334 | 0.153 | 0.203 | 0.132 | 0.231 | 0.406 | 0.280 |
| | | | 192 | 0.335 | 0.378 | 0.207 | 0.252 | 0.150 | 0.249 | 0.417 | 0.284 |
| | | | 336 | 0.362 | 0.403 | 0.250 | 0.283 | 0.162 | 0.261 | 0.430 | 0.290 |
| | | | 720 | 0.393 | 0.428 | 0.319 | 0.330 | 0.194 | 0.286 | 0.458 | 0.308 |
| | | | Avg | 0.340 | 0.386 | 0.232 | 0.267 | 0.160 | 0.257 | 0.428 | 0.290 |
| | PAB | w/o | 96 | 0.269 | 0.333 | 0.148 | 0.195 | 0.135 | 0.230 | 0.402 | 0.274 |
| | | | 192 | 0.333 | 0.378 | 0.192 | 0.240 | 0.151 | 0.244 | 0.433 | 0.287 |
| | | | 336 | 0.358 | 0.401 | 0.242 | 0.276 | 0.169 | 0.262 | 0.443 | 0.300 |
| | | | 720 | 0.391 | 0.428 | 0.315 | 0.328 | 0.207 | 0.301 | 0.473 | 0.318 |
| | | | Avg | 0.338 | 0.385 | 0.224 | 0.260 | 0.165 | 0.259 | 0.438 | 0.295 |

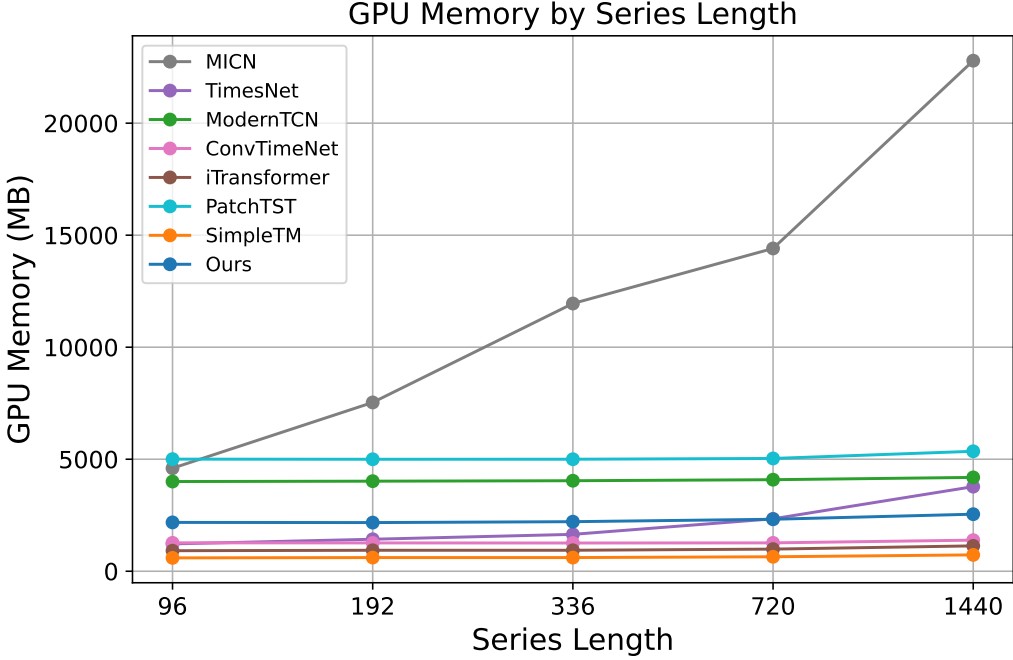

(a) Memory Efficiency Analysis

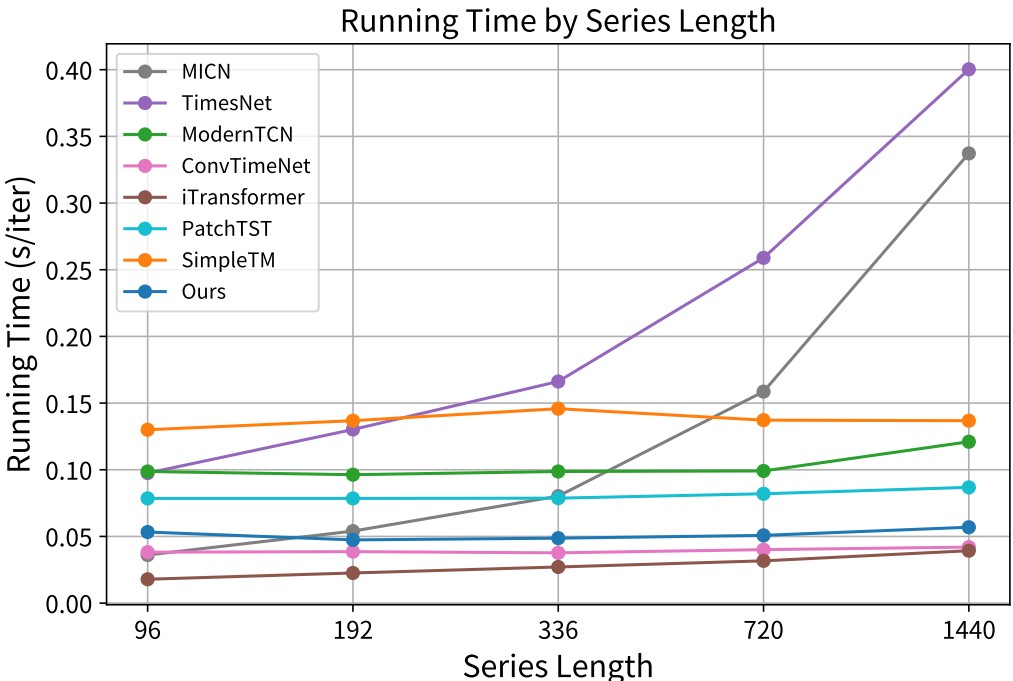

(b) Running Time Efficiency Analysis

Figure 11: Analysis of memory usage and time efficiency of the model on the Weather dataset.

## F USE OF LARGE LANGUAGE MODELS

In preparing this paper we used large language models to assist with writing. They were employed only for language refinement, including grammatical correction and phrasing optimization.

