# OpenReview forum: "TCAN: An Asymmetry Modeling Network for Time Series Forecasting"
_ICLR.cc/2026/Conference — Submitted to ICLR 2026_

### Official Review · Reviewer_AGsG · 2025-10-15

**Soundness:** 2
**Presentation:** 1
**Contribution:** 2
**Rating:** 4
**Confidence:** 4

**Summary:**

This paper introduces a machine learning module, TCAB, which aims to address the research problem where the seasonality pattern among different variables could be different, thus shouldn't be considered to be modelled with shared encoders. The proposed module then builds the proposed model, TCAN.

**Strengths:**

**S1:** The model proposes a plug-and-play module TCAB, as well as a machine learning model TCAN. This enhances the method's general applicability.

**Weaknesses:**

**W1:** The limitation of prior work claimed by the paper is questionable. Sharing an encoder does not mean that the same periodicity is carried across all variables. The paper could benefit from a great revision of its motivation and a more accurate discussion of the limitations in prior work.

**W2:** Although in real life, the unaligned periodicity could (and should) exist, the paper unfortunately shows weak periodicity evidence. The presented periodicities are actually attention weight patterns, not ground truth periodic characteristics. The paper could be improved with a more solid proof that the identified research problem is actually a problem.

**W3:** Experiments are missing for a thorough analysis. See Q2-5.

Minor

- Multiple typos in the paper that need extensive review. For example, in Figure 10, kernel is misspelled as kemerl. Group = M, but in the image it is labelled as D.

- The paper claims TCAB is "mathematically equivalent to patch-wise attention" but in essence, it uses an unnormalized weight that does not sum up to 1. The mathematical rigor could be improved.

- A lot of figures / tables have incomplete captions. The readers are forced to go back to the main paper to really understand what the content is about.

**Questions:**

**Q1:** The paper claims that TCAN can capture the causal relationships, but where does the causality come from? There is no treatment nor graphs, or intervention analysis throughout the process.

**Q2:** Throughout the paper, the authors have been trying to promote the performance gain from using TCAB. However, the presented experiments only show that TCAN works as a better model compared to the baselines. How does the TCAB module work in baseline models? Does it also lead to improved performance?

**Q3:** Why are some datasets (e.g., Solar) only compared with 1 dataset even if the results are provided in the appendix, and there should be no concern in space? Can the authors provide the full comparison?

**Q4:** The "group" setup seems to be important here, as it decides the reception field of how many variables the model can see. However, an analysis of the influence of the number of groups is missing. How does the number of groups affect the model performance in general? How is the parameter decided?

**Q5:** The results are averaged across the full forecasting sequence, i.e., [T+1,T+H]. However, this might over-credit the models that are giving better performance on time steps close to t+1, but bad performance when close to t+H, compared to models that do an average job across all time steps. How is TCAN compared to baselines when only evaluated on the target forecasting horizon t+H?

---

### Official Review · Reviewer_zo8i · 2025-10-21

**Soundness:** 2
**Presentation:** 1
**Contribution:** 2
**Rating:** 2
**Confidence:** 4

**Summary:**

This paper proposes a new convolutional block, called the Temporal Convolutional Association Block (TCAB), and a new framework, called the Temporal Convolutional Association Network (TCAN), to solve the problem of asymmetric modeling in real-world data for time series forecasting. The TCAB comprises a patch-wise association block and a variable-wise association block. The framework can capture asymmetric temporal and causal relationships. Experiments show that the proposed method outperforms most of the selected baselines.

**Strengths:**

1.The structure of this paper is well-organized.\
2.The idea of the method is new, and it outperforms most of the baselines

**Weaknesses:**

1.There some statements in the paper mention that the proposed network can capture causal relationships. However, the paragraph describing the VAB starting at line 270 does not make it clear how it achieves this. And it is not explained how the VAB is an improvement over the original causal convolution used in TCN. How does the VAB better than the origin causal convilution in TCN from the view of cauturing causal relationships? \
2.The motivation for this paper is unclear. While the introduction states that the work builds on insights from attention mechanisms in time-series TSF, its key selling point is better causal relationship capture. The logical connection between these two ideas is not well-established, making the overall argument in the introduction unclear. \
3.In the related work section, only one paper is mentioned that related to causal convolution. To better matching the statement in contribution part, it is suggested to add more papers that focus on improving TCNs from a causal relationship perspective.

**Questions:**

See above

---

### Official Review · Reviewer_Dsc4 · 2025-10-29

**Soundness:** 2
**Presentation:** 3
**Contribution:** 3
**Rating:** 4
**Confidence:** 4

**Summary:**

This paper proposes TCAN, a novel network for multivariate time series forecasting. The model explicitly models asymmetric dependencies, which the authors argue are often missed by standard symmetric encoders. The architecture separates dependencies into two streams: patch-wise (temporal) and variable-wise (cross-channel). This asymmetric approach, in contrast to using a single shared encoder, aims to better capture the distinct nature of temporal and inter-variable relationships.

**Strengths:**

- The paper clearly identifies and motivates modeling asymmetry (both temporal and cross-variable) in time series, an aspect often overlooked by standard symmetric encoders.

- The architecture separates concerns cleanly, with distinct Patch-wise (PAB) and Variable-wise (VAB) blocks that lend themselves well to component-wise ablation studies.

- Ablation studies successfully isolate the contributions of key architectural components, showing clear benefits from asymmetry modeling.

**Weaknesses:**

- Key details remain underspecified, such as how the convolutional blocks are "equivalent to patch-wise attention logits." This equivalence claim is unsubstantiated and should be formally proven or rephrased as an analogy.

- The paper's central premise is modeling asymmetry, yet it provides no diagnostic analysis, visualization, or interpretation of actual learned asymmetry patterns. Please add visualizations of the learned TCAB filter weights to provide insight.

- The claim that "most existing methods ignore asymmetry" overstates the novelty, as prior work (e.g., RNNs, causal models, GNNs) handles directionality. The authors should tone down this claim.

- Main results lack standard deviations or confidence intervals, even though multiple runs were reportedly performed.

**Questions:**

- Can you provide a formal proof or specify the precise conditions under which the grouped convolution mechanism (TCAB) is "mathematically equivalent to patch-wise attention logits"?

- How are the convolutional filters (in TCAB) initialized and constrained? Are they shared across layers? Is any regularization used?

- Can you provide parameter count and memory/runtime analysis to quantify the overhead of the asymmetric (grouped) parameterization as the number of variables (M) grows?

- Can you resolve the inconsistency regarding TCAB/PAB kernel sizes? Are they 1x1 as stated in the text, or (P,1) as suggested in the figures? Please provide a canonical shape table.

- Have you considered GNN-based forecasting methods and classical statistical models, which also handle directional dependencies?

---

### Official Review · Reviewer_Tf2K · 2025-10-30

**Soundness:** 2
**Presentation:** 2
**Contribution:** 2
**Rating:** 4
**Confidence:** 4

**Summary:**

The paper focuses on multivariate time series forecasting and claims that most existing methods implicitly assume symmetric modeling, i.e., sharing the same encoder across all variables under the assumption of periodic consistency. The authors argue that real-world variables exhibit heterogeneous periodicities and propose an asymmetric modeling strategy implemented via a new module called the Temporal Convolutional Association Block (TCAB). TCAB combines group convolution and attention-like mechanisms to model “patch-wise” and “variable-wise” dependencies with separate parameters. The overall architecture, TCAN, reportedly achieves SOTA performance on seven datasets.

**Strengths:**

1. TCAB bridges attention and convolution in a principled way, maintaining attention’s expressive power while leveraging convolution’s efficiency and positional awareness.
2. The proposed TCAN achieves SOTA performance across seven real-world datasets, indicating robustness and generality.

**Weaknesses:**

1. The paper’s focus is unclear. Is it addressing the statistical inconsistency of variable periodicity or the computational inefficiency of attention? The motivation feels mixed.
2. The motivation for the method is weak. Many prior works (e.g., channel-independent models, graph-based TSF, variable-wise attention) have already handled heterogeneous variables. The claim of “first to reveal symmetric modeling” doesn’t really hold . It sounds like the authors just renamed the old “parameter sharing” issue.
3. The experiments are not convincing. The baselines only include CNN- and Transformer-based models, while stronger and lighter MLP-based methods are missing. The performance gain is around 1%, which doesn’t clearly show the method’s advantage or prove that it actually solves the claimed “variable periodicity inconsistency” problem.

**Questions:**

1. Please report multi-run results (mean and std) to demonstrate the robustness of the proposed method.
2. Include more baselines, especially recent MLP-based TSF models, to better support the claimed performance improvement.
3. Provide stronger evidence that the proposed “variable periodicity inconsistency” problem is fundamentally different from existing works, and show via experiments that the proposed model indeed addresses this issue.

---

### Meta-Review · Area_Chair_QffK · 2025-12-17

**Summary:**

This paper presents TCAN, a method combining attention and convolution to address modeling asymmetry in time series forecasting. Reviewers recognize the well-structured architecture and its strong empirical performance. However, the consensus identifies major concerns that must be resolved. The core motivation regarding "variable periodicity inconsistency" is seen as unclear and its novelty overstated, as it resembles known parameter-sharing challenges. The experiments are insufficient, lacking comparisons to strong MLP-based baselines, robust statistical reporting, and ablation studies that isolate the benefit of the proposed TCAB module. Furthermore, claims of capturing "causal relationships" are unsupported, key technical details (e.g., mathematical equivalence proofs, filter initialization) are missing, and the presentation requires careful proofreading. I think substantial revisions are needed to clarify the foundational problem, rigorously validate claims with expanded experiments and analysis, and improve the technical depth and clarity of the writing.

**Reviewer Concerns:**

There is no rebuttal from the authors such that all concerns are still outstanding.

**Reviewer Scores:**

There is no discussion between the reviewers and the authors such that the scores remain the same as orginal.

---

### Decision · Program_Chairs · 2026-01-26

Reject